# Desilication of Sodium Aluminate Solutions from the Alkaline Leaching of Calcium-Aluminate Slags

**James Malumbo Mwase \* and Jafar Safarian** [ID]

Department of Materials Science and Engineering, Norwegian University of Science and Technology (NTNU), 7491 Trondheim, Norway
\* Correspondence: james.mwase@ntnu.no

**Abstract:** The desilication of sodium aluminate solutions prior to precipitation of aluminum tri-hydroxides is an essential step in the production of high purity alumina for aluminum production. This study evaluates the desilication of sodium aluminate solutions derived from the leaching of calcium-aluminate slags with sodium carbonate, using CaO, Ca(OH)$_2$, and MgO fine particles. The influence of the amount of CaO used, temperature, and comparisons with Ca(OH)$_2$ and MgO were explored. Laboratory scale test work showed that the optimal conditions for this process were using 6 g/L of CaO at 90 °C for 90 min. This resulted in 92% of the Si being removed with as little as 7% co-precipitation of Al. The other desilicating agents, namely Ca(OH)$_2$ and MgO, also proved effective in removing Si but at slower rates and higher amounts of Al co-precipitated. The characteristics of solid residue obtained after the process indicated that the desilication is via the formation of hydrogarnet, Grossular, and hydrotalcite dominant phases for CaO, Ca(OH)$_2$ and MgO agents, respectively.

**Keywords:** desilication; silica; pedersen process; CaO





## 1. Introduction

Desilication of sodium aluminate solutions is an essential step in the production of alumina through the Bayer process. In this process, bauxite ores containing silicon are leached in an alkaline media, with the primary purpose of extracting aluminum. However, silicon is often co-extracted due to a reaction with sodium hydroxide (Equation (1)), which can contaminate the final alumina product. To prevent this, a desilication process to reduce the amount of silicon in solution is conducted prior to precipitating hydrated alumina. In the Bayer process, bauxite ores are pressure leached at a high temperature (100–250 °C) using sodium hydroxide solution. The leachate solution is then cooled and seeded to precipitate alumina hydrates. Desilication of this leachate prior to precipitation is achieved through the addition of CaO solid particles in the leaching phase. This also aids in the regulation of carbonates and phosphates, which in high concentrations are detrimental to the precipitation process. Further, the presence of CaO accelerates the leaching of aluminum when it is in the mineral phase diaspore, which is the most difficult alumina mineral to leach. The chemistry of Si during the desilication has been described by a few studies [1–3] as follows.

$$SiO_{2(s)} + 2NaOH = Na_2SiO_{3(aq)} + H_2O \tag{1}$$

The soluble products formed in leaching, namely NaAlO$_2$ and Na$_2$SiO$_3$, react to form non-soluble aluminosilicate precipitates with zeolite structures and are termed desilication products (DSP) of Na$_2$O.Al$_2$O$_3$.2SiO$_2$ or Na$_8$Al$_6$Si$_6$O$_{24}$(OH)$_2$. These DSPs further react with sodium hydroxide and carbonates in the solution to form sodalite (Na$_8$Al$_6$Si$_6$O$_{24}$(CO$_3$).2H$_2$O). The whole process can be considered a 'self-desilication'. The addition of CaO results in the rest of the Si reacting to form cancrinite (Na$_6$Ca$_2$Al$_6$Si$_6$O$_{24}$(CO$_3$)$_2$.2H$_2$O), which is a slightly more soluble phase.

A recent study investigated the use of the Pedersen process as an alternative to the Bayer process [4]. The study also evaluated the process as a means to valorize solid waste residues from the Bayer process such as bauxite residue (red mud). In the investigation calcium aluminate slags obtained via a prior smelting-reduction process, were leached with sodium carbonate solution to extract alumina. The solution was then sparged with $CO_2$ gas to precipitate hydrated alumina (carbonation). Unlike the Bayer process where CaO is added during leaching, the study [4] added it after leaching, adapting the method used in a similar investigation [5]. It was observed that without the desilication step, the alumina precipitated had a high silica content exceeding the standard for metallurgical grade alumina [4]. Both studies [4,5] used this method at laboratory testing stage. It is speculated that if added during the leaching phase of the Pedersen process, the CaO may be consumed to form $CaCO_3$ (grey mud) rather than desilication products. Before carbonating with $CO_2$ gas to precipitate hydrated alumina the leachate solution was desilicated using up to 28 g of powdered CaO per 1 L of leachate solution at a temperature of 70 °C. The process successfully reduced the amount of silicon in solution from concentrations of 0.4 g/L to 0.02 g/L [4]. Considerable Al co-precipitation, ranging from 18 to 50%, was also observed. The process yielded a product that was mostly calcium monocarboaluminate (50–70%), followed by calcite (11–36%), portlandite (6–10%), and hydrogarnite (1–18%). It was concluded that the desilication mostly occurred via the formation of hydrogarnite ($Ca_3Al_2(SiO_4)_{3-x}(OH)_{4x}$). Prior to using this method, the chemistry and mechanisms of this process were not known and are still not fully understood. Based on the results the authors [4] proposed three possible mechanisms for the process based on the previous studies [6–10].

In the first mechanism Si in the form of $H_7SiO_6^-$ reacts with CaO to form hydrogarnet by the reaction:

$$3Ca^{2+} + 2AlO_2^- + (3-x)H_7SiO_6^- + (x+1)OH^- = Ca_3Al_2(SiO_4)_{3-x}(OH)_{4x} + (11-5x)H_2O \tag{2}$$

In a second mechanism Si is in the form of $HSiO_3^-$ reacting to form hydrogarnet by the reaction:

$$3Ca^{2+} + 2AlO_2^- + (3-x)HSiO_3^- + (x+1)OH^- + (2x-1)H_2O = Ca_3Al_2(SiO_4)_{3-x}(OH)_{4x} \tag{3}$$

It was further proposed that Si was removed from the solution by the formation of calcium monocarboaluminate, which in turn reacted with the Si in solution forming hydrogarnite. In some tests there was no formation of hydrogarnet, despite up to 27% of the Si being removed. In these cases, it is proposed that the monocarboaluminate may have adsorbed the Si ions and not necessarily reacted with them. Calcium monocarboaluminate is most likely the result of a complex reaction between the CaO, unreacted sodium carbonate and the newly formed sodium aluminate complex from leaching.

The other products of note, namely calcite and portlandite were possibly formed by reactions identical or similar to the causticisation reactions described by [11]. Calcium oxide first reacts with water to give portlandite (Equation (4)).

$$CaO_{(s)} + H_2O \rightarrow Ca(OH)_{2(s)} \tag{4}$$

Some of the portlandite reacts with unreacted sodium carbonate to give calcite (Equation (5)).

$$Ca(OH)_{2(s)} + CO_3^{2-} \rightarrow CaCO_{3(s)} + 2OH^- \tag{5}$$

The present study aims to optimize the use of the desilication process as part of the Pedersen process. The influence of the amount of CaO used, temperature, and comparisons with other desilication agents (MgO and $Ca(OH)_2$) will be explored.

## 2. Materials and Methods

### 2.1. Solution Preparation

The solutions for the desilication tests were obtained by leaching a 100 g calcium aluminate slag sample (Table 1) with 1 L of 60 g/L of sodium carbonate solution at a temperature of 90 °C for a duration of 90 min. It is worth noting that the slag was produced via the smelting-reduction of bauxite with lime and carbon. On completion of each leaching test the slurry was vacuum filtered, recovering around 950 mL of the solution, which was allowed to cool overnight before the desilication tests.

**Table 1.** XRF analysis of the calcium aluminate slag.

| $Al_2O_3$ | CaO | $SiO_2$ | $TiO_2$ | MgO | Fe | $SO_3$ | $P_2O_5$ |
|---|---|---|---|---|---|---|---|
| 43.90 | 52.12 | 2.00 | 1.30 | 0.44 | 0.25 | 0.07 | 0.01 |

### 2.2. Desilication and Analyses

Before each test commenced, a sample of solution was taken to determine the amount of Si, Al, and Ca at the start of each test. The experiments were conducted in a jacketed glass reactor. The mixture was heated using silicon oil circulated using a VWR thermobath circulator. The temperature of the mixture was measured using a temperature probe connected to the circulator. The mixture was agitated using an overhead stirrer with a shaft and paddle impeller. Figure 1 shows a drawing of the experimental set-up. All the tests were conducted by adding the leachate solution to analytical reagent grade powdered CaO in the glass reactor. In each test, the mixture was heated to the desired temperature while being agitated at 400 rpm for 90 min. The time it took to reach the desired temperature was included in the entire duration of each experiment (90 min), and it varied according to the set temperature. Samples of solution were withdrawn at 15 min intervals for the duration of the test and vacuum filtered with 0.22 μm membrane for ICP-MS analysis of Si, Al, and Ca. At the end of each test, the reactor was emptied via a drainage valve at the bottom of the vessel. The contents were vacuum filtered with general purpose Whatman filter paper. The filter cake was dried at 60 °C for 2 days and analyzed via XRD. The amount of CaO used varied between 1 and 12 g. Three different temperatures were used, namely 40, 70, and 90 °C.

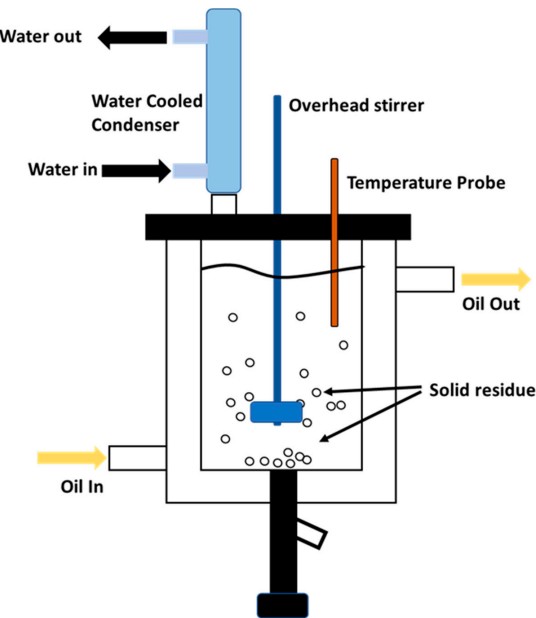

**Figure 1.** Schematic drawing of experimental equipment.

Further, comparison was made with the results from using Ca(OH)$_2$ and MgO as the desilication agent as opposed to CaO.

## 3. Results

This section may be divided by subheadings. It should provide a concise and precise description of the experimental results, their interpretation, as well as the experimental conclusions that can be drawn.

### 3.1. ICP-MS Analysis of Desilicated Solutions

The results from the analyses of solution samples and the solid residues are detailed below and discussed in the next section. The results for the tests using 1 and 2 g of CaO are excluded, as those amounts were ineffective and did not remove any Si. Table 2 shows the percentage reduction of Si with time and in total how much Si was removed. Moreover, Table 3 shows the reduction of Al concentration in g/L with time and in total how much Al was co-precipitated or lost at the end of the test.

**Table 2.** The desilication conditions and percentage reduction of Si with time.

| Test No. | 1 | 2 | 3 | 4 | 5 | 6 | 7 | 8 | 9 |
|---|---|---|---|---|---|---|---|---|---|
| CaO added (g) | 12 | 6 | 4 | 6 | 6 | 4 | 4 | | |
| Ca(OH)$_2$ added (g) | | | | | | | | 6 | |
| MgO added (g) | | | | | | | | | 6 |
| Temp (°C) | 90 | 90 | 70 | 40 | 70 | 40 | 90 | 90 | 90 |
| Initial Si concn. (g/L) | 0.3 | 0.2 | 0.2 | 0.2 | 0.2 | 0.2 | 0.2 | 0.2 | 0.2 |
| **Time (mins)** | | | | | | | | | |
| 0 | 100 | 100 | 100 | 100 | 100 | 100 | 100 | 100 | 100 |
| 15 | 21 | 77 | 74 | 66 | 81 | 71 | 68 | 74 | 58 |
| 30 | 19 | 59 | 67 | 64 | 68 | 65 | 69 | 45 | 52 |
| 45 | 21 | 31 | 51 | 61 | 59 | 63 | 61 | 26 | 40 |
| 60 | 17 | 16 | 44 | 53 | 52 | 62 | 40 | 15 | 29 |
| 75 | 15 | 9 | 33 | 47 | 49 | 58 | 58 | 10 | 27 |
| 90 | 13 | 9 | 25 | 34 | 49 | 52 | 61 | 8 | 25 |
| **% of Si removed** | 87 | 91 | 75 | 66 | 51 | 48 | 39 | 92 | 75 |

**Table 3.** Reduction in Al concentration (g/L) with time.

| Test No. Time (mins) | 1 | 2 | 3 | 4 | 5 | 6 | 7 | 8 | 9 |
|---|---|---|---|---|---|---|---|---|---|
| 0 | 16.4 | 15.1 | 13.7 | 13.3 | 12.6 | 14.4 | 13.3 | 14.8 | 13.4 |
| 15 | 14.3 | 15.0 | 12.1 | 10.9 | 12.4 | 12.8 | 12.3 | 13.3 | 10.6 |
| 30 | 12.6 | 14.6 | 12.7 | 11.9 | 12.5 | 12.8 | 12.4 | 13.0 | 12.0 |
| 45 | 13.5 | 13.5 | 13.1 | 12.6 | 12.5 | 12.8 | 11.9 | 13.3 | 11.4 |
| 60 | 13.0 | 13.6 | 13.1 | 12.6 | 12.3 | 13.3 | 8.9 | 12.9 | 9.9 |
| 75 | 13.5 | 14.1 | 13.0 | 12.5 | 12.0 | 13.5 | 12.0 | 12.8 | 10.7 |
| 90 | 13.4 | 14.1 | 12.8 | 11.1 | 11.6 | 12.6 | 12.8 | 12.5 | 10.3 |
| **% of Al co-precipitated** | 18.3 | 6.6 | 3.2 | 16.1 | 8.5 | 12.5 | 3.2 | 15.4 | 22.9 |

### 3.2. XRD Analysis of Solid Residue

Table 4 shows the various phases present in the solid residue recovered after each desilication test. The samples were analyzed using a Bruker D8 A25 DaVinci™ X-ray Diffractometer with CuKα radiation. The identification and quantification of phases in all residue samples were carried out by the Rietveld method using EVA and TOPAS software respectively. Obviously, silicon removal has occurred through the formation of some Si-containing phases, while Al has been partially lost to the precipitate. Figure 2 shows a

typical XRD spectrum for test 2 solid residue with the identified phases. A few peaks, minor phases, such as the one at 11.5 degrees, were not unfortunately identified for this sample, while for the others they were most identified.

**Table 4.** XRD analysis quantifying mineral phases by wt% in solid residue.

| Test No. / Mineral Phase | 1 | 2 | 3 | 4 | 5 | 6 | 7 | 8 | 9 |
|---|---|---|---|---|---|---|---|---|---|
| Hydrogarnet [1] | 71.4 | 80.0 | 47.3 | 47.8 | 1.0 | 22.0 | 7.0 | 28.0 | |
| Calcium monocarboaluminate [2] | 24.4 | | 29.7 | 30.8 | 63.0 | 60.0 | 21.0 | | |
| Portlandite (Ca(OH)$_2$) | 4.0 | 8.0 | 5.2 | 4.9 | 30.0 | 11.0 | 5.0 | 4.0 | |
| Calcite (CaCO$_3$) | 0.2 | 10.0 | 2.5 | 2.7 | 6.1 | 0.5 | 53.0 | 1.0 | |
| Hydrotalcite [3] | | 2.0 | | | | | | 1.0 | 64.0 |
| Strätlingite [4] | | | 15.3 | 13.8 | | 6.5 | 14.0 | | |
| Grossular [5] | | | | | | | | 64.0 | |
| MgO | | | | | | | | | 36.0 |

[1] $Ca_3Al_2(SiO4)_{3-x}(OH)_{4x}$ [x = 1.5–3]. [2] $Ca_4Al_2(OH)12CO_3.5H_2O$. [3] $Mg_4Al_2(CO_3)(OH)_{12}.3H_2O$. [4] $Ca_2Al(AlSi)_2O_2(OH)_{10}$. [5] Less crystalline hydrogarnet.

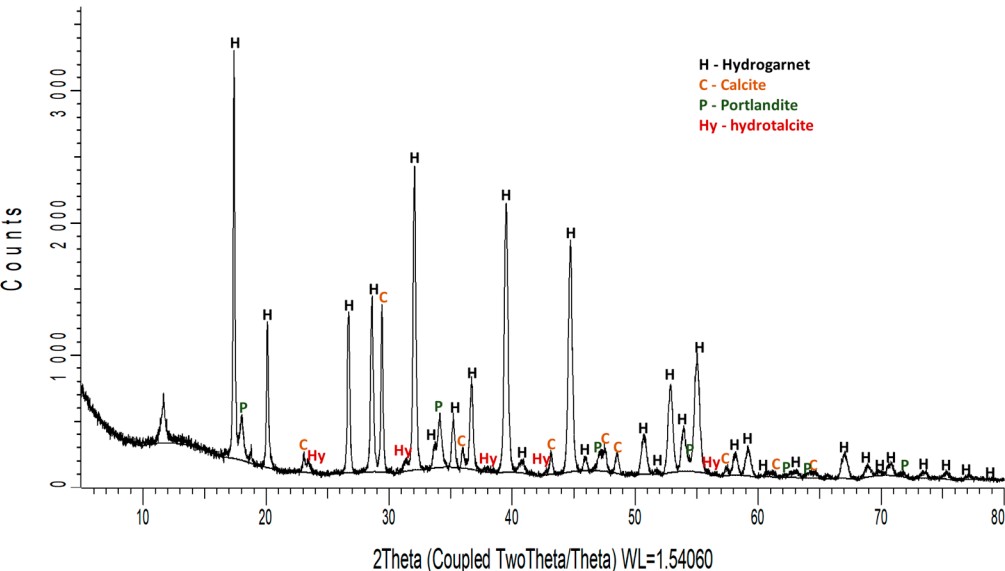

**Figure 2.** XRD spectrum for test 2 residue.

## 4. Discussion

### 4.1. Effect of Added CaO Mass

An initial comparison is made between tests 1, 2, and 7. These three tests were conducted at 90 °C using 12, 6, and 4 g of CaO. The results in percentage reduction of Si over time can be seen in Figure 3. It should be noted that Figures 3–6 show the reduction of silica in solution as a percentage of the original amount before desilication commenced. Tests 1 and 2 had similar amounts of Si removed, with the test 2 removing a bit more than test 1. In test 2, the amount of Si was reduced to 9% of the starting amount. This means CaO effectively removed 91% of the Si. Test 7 lagged behind the other two, only removing 39% of the Si. Further, tests 1 and 2 removed comparable amounts of Si to the 28 g tests conducted by [12]. This shows that the desilication can be successfully conducted using significantly lower CaO per volume of leachate solution that we experienced before. Comparing the two tests shows that the amount of CaO present clearly influenced the kinetics of the reaction. In test 1, the Si is mostly removed in a period of 20 min, while test 2 required 60 min to remove an equal amount of Si. The XRD analysis results of the solid residue (Table 4) indicate a larger presence of hydrogarnet in tests 1 and 2 than was observed in the tests conducted

by [12] where 28 g of CaO was used. In those tests only up to 12% of the residue was hydrogarnet with the majority being calcium monocarboaluminate followed by portlandite and calcite. This means that the hydrogarnet is the more thermodynamically stable and favored product, reactions (2) and (3), and once equilibrium is reached the remaining or excess CaO reacts with unreacted $NaCO_3$ to form calcium monocarboaluminate. As before, it is proposed that the Si removal occurred due to the formation of hydrogarnet and possibly by adsorption of Si ions to the multi-layered double hydroxides in the calcium monocarboaluminate also present in tests 1 and 7. The mineral phase profile of the solid residue from test 7 is curious in that it contains Strätlingte. This phase has not been seen in previous results. From its formula it is clear it is responsible for some Si removal as well as some co-precipitation of Al. As expected, co-precipitation of Al occurred, and Table 3 shows the reduction in concentration with time and the final amount of Al lost as a percentage of the starting amount at time 0.

Similar to test 7 (4g), there were several tests whose results had data points that appeared to be Si gains. These cases are most likely Si being re-dissolved back into solution, the reason for which is unknown. This may be the Si that is not chemically bonded and part of the hydrogarnet compound, but that which has been adsorbed by the monocarboaluminate.

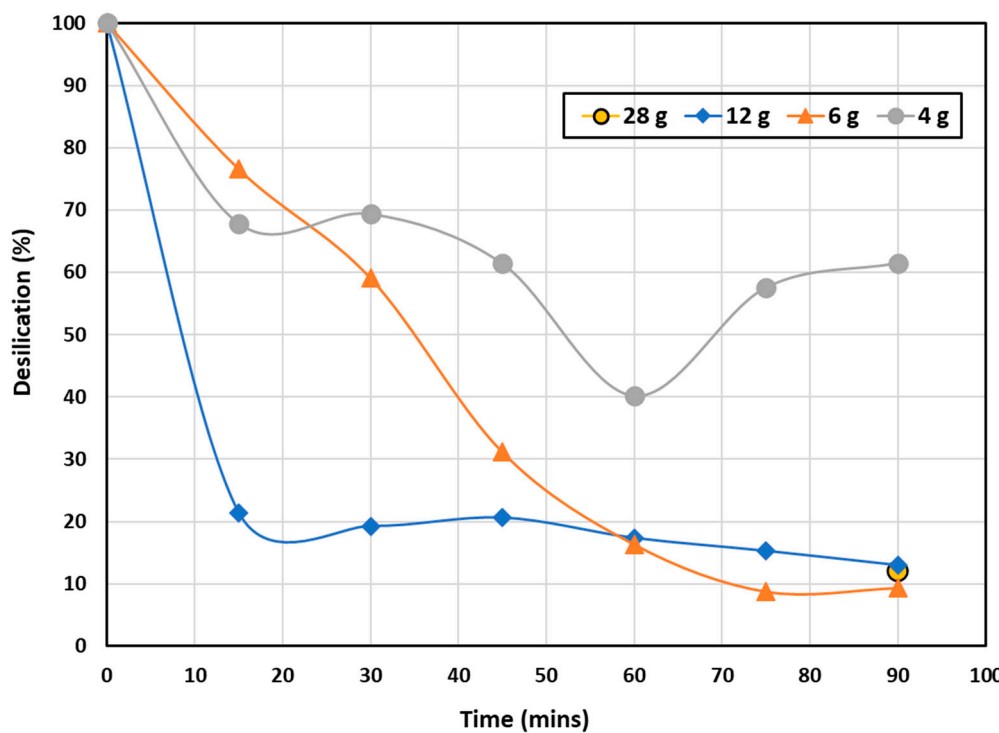

**Figure 3.** Percentage reduction of Si with time using 28 g [12], and tests 1, 2, and 7 with 12, 6, and 4 g of CaO at 90 °C for 90 min.

### 4.2. Effect of Temperature

Further tests were conducted with 6 and 4 g of CaO added and the influence of temperatures of 70 and 40 °C were noted and compared with the 90 °C tests. Figure 4 shows that the most effective temperature for desilication using 6 g of CaO is 90 °C (test 2). Test 4 and 5 performed similarly, although given test 5 had an operating temperature closer to test 2, it was expected it would remove an amount of Si closer to this test and not less than test 4. Table 4 shows that the solid residue of test 2 was significantly different from test 4 and 5. The higher temperature of 90 °C favored the formation of hydrogarnet over calcium monocarboaluminate, which formed more readily at 70 and 40 °C. This perhaps also explains why test 2 had markedly better Si removal that test 4 and 5. The removal of

about 50% of Si in test 5 and the formation of calcium monocarboaluminate and portlandite as the dominant phases in the residue may indicate that these phases have Si impurity in their structure.

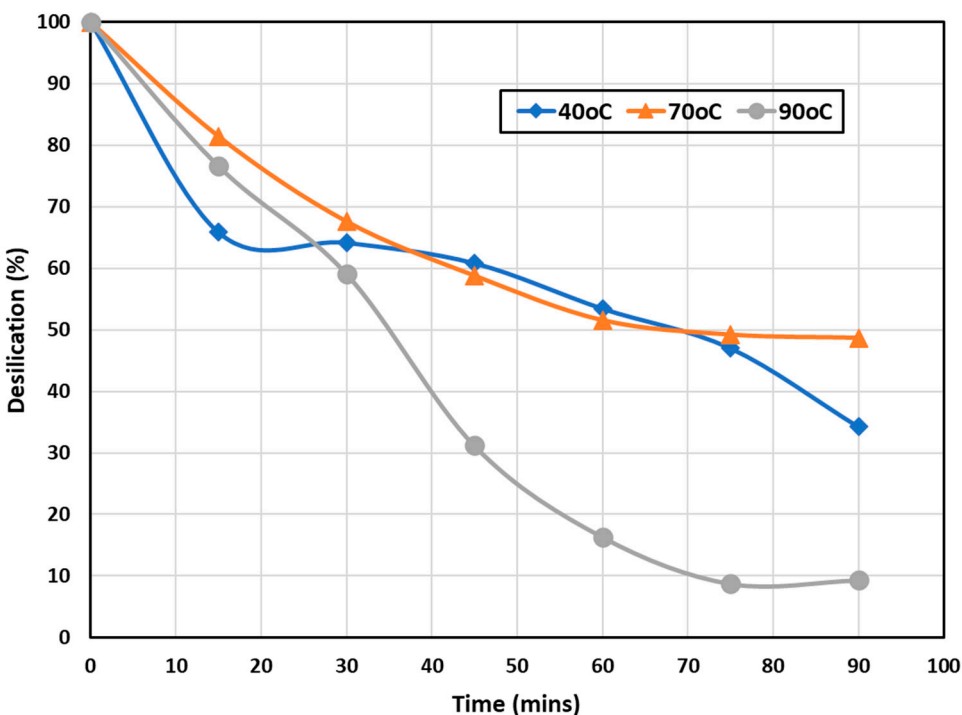

**Figure 4.** Percentage reduction of Si with time using 6 g CaO at temperatures 40 °C (Test 4), 70 °C (Test 5), and 90 °C (Test 2).

In the case of the tests using 4 g of CaO, Figure 5, the best performing test, was test 3 conducted at 70 °C. This contrasts with the tests using 6 g, where the best performing test was the one conducted at 90 °C. In this case the test 7 at 90 °C performed poorly, and extracted even less Si than test 6 at 40 °C. Table 4 shows that the mineral phase composition was qualitatively similar across the three tests with variation in the amounts. As already seen the test with the highest amount of hydrogarnet (test 3) has the highest Si removal. Test 7 with the lowest amount of hydrogarnet and highest amount of calcium monocarboaluminate. The formation of strätlingite phase in these three tests and formation of almost the same amounts of this phase in tests 3 and 7 may strengthen the idea that calcium-monocarboaluminate can potentially contain some Si impurity.

Considering the different results at various temperatures with different amounts of CaO added may be related to the equilibrium in the system and the sequence of reactions. For the best result obtained, test 2 at 90 °C, we observe hydrogarnet as the dominant Si-containing phase. However, for the test 7 at the same conditions with less CaO addition, hydrogarnet is not the dominant phase, while calcite, calcium monocarboaluminate, and strätlingite are dominant phases, respectively. It may be possible that more CaO addition into the system at 90 °C is accompanied with the conversion of strätlingite to hydrogarnet. Hence, with CaO addition at 90 °C, we may have the first calcite formation via chemical reactions (4) and (5), and simultaneously strätlingite precipitation. If CaO addition is added more, more Ca is available and hydrogarnet is formed.

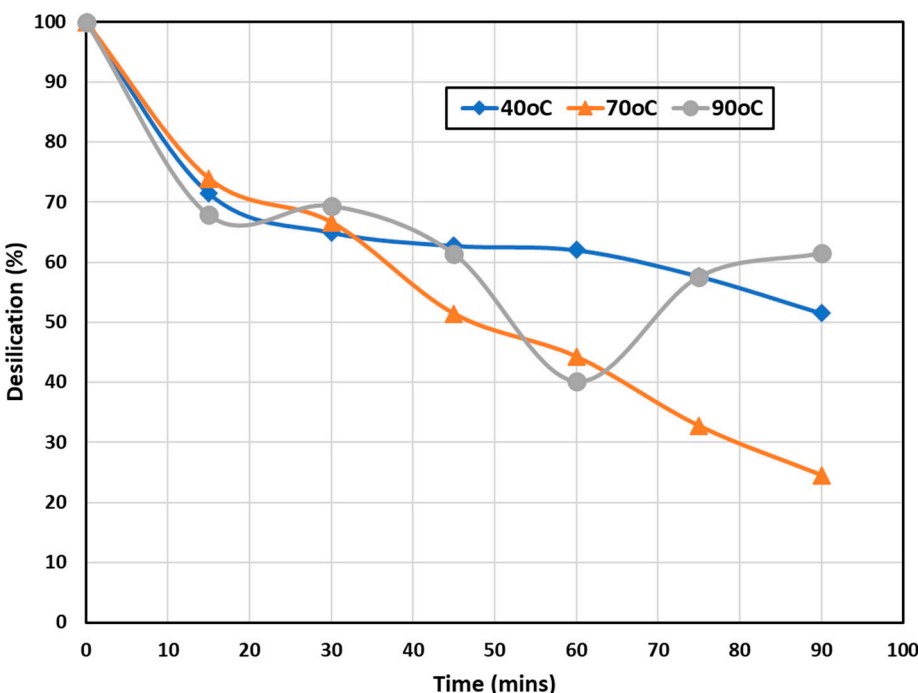

**Figure 5.** Percentage reduction of Si with time using 4 g CaO at temperatures 40 °C (Test 6), 70 °C (Test 3), and 90 °C (Test 7).

### 4.3. Effect of Type of Desilication Agent

Figure 6 shows a comparison and results when using different desilicating agents under identical conditions. Overall, $Ca(OH)_2$ removed the most Si, equaling 93% (Table 2). However, this is only 1% more than CaO. Table 4 shows the product from $Ca(OH)_2$ desilication was mostly hydrogarnet. Specifically, two types of hydrogarnet differ in the degree of crystallinity. The mechanism of hydrogarnet formation using $Ca(OH)_2$ is most likely identical to the ones proposed for CaO in Section 1 via chemical reactions (2) and (3). Although MgO removed the least amount (75%), it is worth noting it removed a considerable amount of Si. The major product in this case was hydrotalcite. Based on the chemical formula, it is assumed that this formed due to a complex reaction between the MgO, $NaAlO_2$, and unreacted sodium carbonate. However, an attempt to balance this equation and rewrite it in different ionic forms using HSC Chemistry software showed that the reaction was not possible. It is therefore assumed that the reaction takes place in a series of reactions forming intermediate products before the hydrotalcite is formed. Hydrotalcite is a layered double hydroxide [13]. Hence, the Si removal in the case of the MgO is probably due to the Si ions being adsorbed in between these double layers. This accounts for the Si removal in this test despite none of the products having Si in their chemical formulas.

The different rates of reaction are also quite noticeable among the different desilicating agents. CaO, as noted earlier, has a very short reaction time to achieve maximum Si removal, while $Ca(OH)_2$ is more gradual taking three times the amount of time to achieve the same result. Deciding the best desilicating agent will also take into consideration the amount of Al co-precipitated. It can be seen in Table 3 that CaO experiences the least amount of co-precipitation (6.7%). Although $Ca(OH)_2$ is just as effective as CaO in removing Si, it co-precipitates more than double the amount (15%) of Al as CaO. MgO is completely discounted, as it co-precipitates 23% of the Al.

Comparing the characteristics of the resulting residues from the three different desilication agents, it is observed that when CaO and $Ca(OH)_2$ have been used for 90 min, they were mostly converted to the other products, while for MgO there is a significant amount of MgO unreacted (Table 3), and Figure 6 clearly shows that it may be due to the slower desilication rate and need for longer reaction time for this agent. If MgO has not completely

reacted due to the passivation via the formation of hydrotalcite over MgO particle, the particle size of this agent must be optimized, which is probably not a case for CaO and Ca(OH)$_2$ agents.

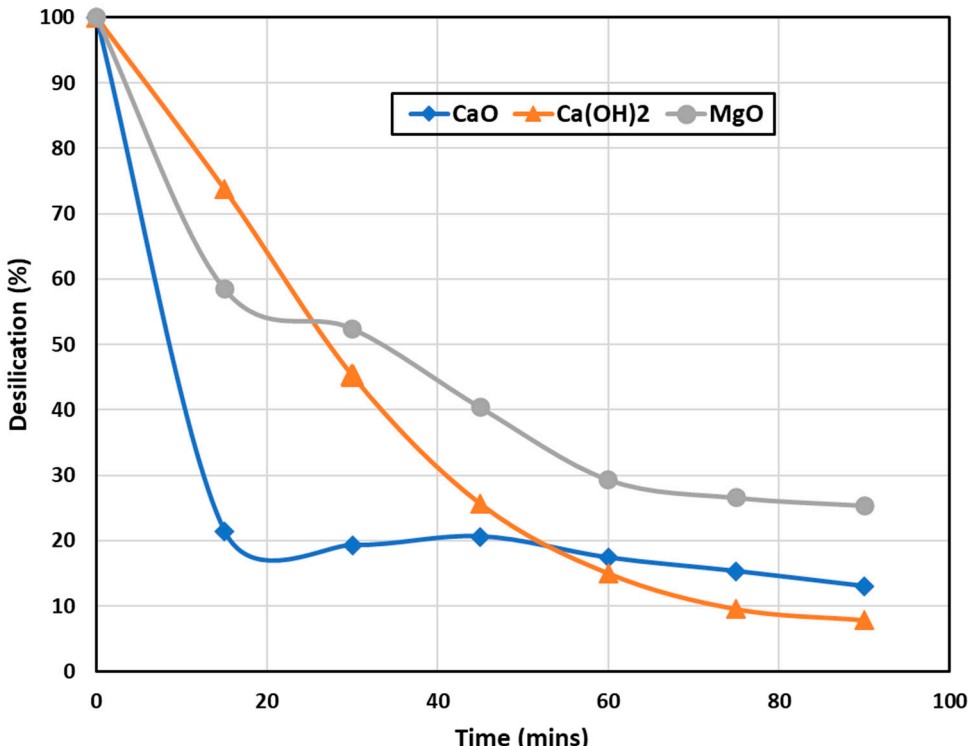

**Figure 6.** Percentage reduction of Si with time using 6 g of CaO, Ca(OH)$_2$, and MgO at 90 °C for 90 min.

*4.4. Characteristics of the Desilicated Solutions*

A comparison is made between the desilication results achieved in the current study and one which investigated adapting the Pedersen process for ferruginous bauxites of the Pacific Northwest in USA [5]. In that study, after several attempts, a leachate with 0.19 g/L of Si was obtained, using comparable leaching parameters to this study. Desilication including a first step using 28 g/L CaO at 90 °C for 1 h, then a second step using 2g/L CaO, resulted in 95% Si removal and up to 27% co-precipitation of Al. This is similar to the results achieved in this study, but with more CaO used.

A comparison with the desilication in the Bayer process is not easily conducted, as the amounts of CaO vary widely depending on the type of ore processed and the leaching temperature. Karstic type bauxites require high processing temperatures and addition of CaO, while Tropical type bauxites require lower leaching temperatures and sometimes no addition of CaO. What this study has established is that in comparison to the other works in this field [4,5], desilication can be achieved with a lower addition of CaO and correspondingly lower Al loss from the solution in this step. It has also shed some light on the mechanisms of the process.

**5. Conclusions**

The desilication of a sodium aluminate solution obtained from calcium aluminate slags was carried out using CaO, Ca(OH)$_2$, and MgO. The concluding remarks are summarized as:

- Calcium oxide has shown to be an effective desilicating agent of leachate solutions, depending on the amount added and applied temperature desilication varies.
- CaO works faster than other desilicating agents tested, namely calcium hydroxide and magnesium oxide, and co-precipitates less alumina.

- Desilication by CaO is mainly via the formation of hydrogarnet, while it is via the formation of both hydrogarnet and grossular by $Ca(OH)_2$. When MgO is used, silicon is removed via hydrotalcite formation.
- The desilication occurs largely due to the formation of hydrogarnet, and to a smaller extent, the adsorption of Si in between the layers of calcium monocarboaluminate, which is a layered double hydroxide. The optimal conditions for this process are 6 g of CaO for 1 L of leachate solution, with 0.3 g/L Si at 90 °C for 90 min.

**Author Contributions:** Conceptualization, J.M.M. and J.S.; methodology, J.M.M. and J.S.; software, J.M.M. and J.S.; validation, J.M.M. and J.S.; formal analysis, J.M.M. and J.S.; investigation, J.M.M.; resources, J.S.; data curation, J.M.M.; writing—original draft preparation, J.M.M.; writing—review and editing, J.M.M. and J.S.; visualization, J.M.M. and J.S.; supervision, J.S.; project administration, J.M.M. and J.S.; funding acquisition, J.S. All authors have read and agreed to the published version of the manuscript.

**Funding:** This project has received funding from the European Union's Horizon 2020 research and innovation programme under grant agreement 767533.

**Institutional Review Board Statement:** Not applicable.

**Informed Consent Statement:** Not applicable.

**Data Availability Statement:** Not applicable.

**Acknowledgments:** We also thank the partners in the ENSUREAL consortium for their continued support.

**Conflicts of Interest:** The authors declare no conflict of interest.

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
