# Peer review of "Desilication of Sodium Aluminate Solutions from the Alkaline Leaching of Calcium-Aluminate Slags"

_processes, doi:10.3390/pr10091769_

Round 1
Reviewer 1 Report
In this paper, the authors tested CaO, Ca(OH)2, and MgO as the desillication agent to remove Si from the calcium-aluminate slags. They studied several parameters like reactant amount, reaction temperature, and reaction time. The optimal processing condition is adding 6g of CaO into 1L of leachate solution at 90C for 90 mins.
Questions:
1. What happened to the content in lines 131-149? Are they supposed to be here? Have you prove-read before submission?
2. In table 2, why the test No.1 has a 0.3 g/l concentration? There is no clear variable control in the experimental design
3. In the desilication results figures, the y-axis is the desilication percentage. Does that mean when reaching 100% there is no Si in the solution?
4. In figure 3,4. and 5, there are several data points showing a positive gain of silicon; for example in figure 3 the gray line at 30C and 75C. What is the reason behind this phenomenon? Is this result repeatable? What is the error bar for each data point?
Author Response
1. What happened to the content in lines 131-149? Are they supposed to be here? Have you prove-read before submission?
Apologies, the content has been deleted.
2. In table 2, why the test No.1 has a 0.3 g/l concentration? There is no clear variable control in the experimental design
The test work used solution from leaching actual slag hence it was impossible to have exactly the same concentrations every time. It was not a synthetic solution in which the concentrations could be made the same with every solution for each test.
3. In the desilication results figures, the y-axis is the desilication percentage. Does that mean when reaching 100% there is no Si in the solution?
No, as explained in the caption of each figure the curve shows the reduction in silica in solution as a percentage of the original amount of silica before desilication commenced. This has now been explained in lines 182-184. And the captions for all Figures have been changed to reflect this information.
4. In figure 3,4. and 5, there are several data points showing a positive gain of silicon; for example in figure 3 the gray line at 30C and 75C. What is the reason behind this phenomenon? Is this result repeatable? What is the error bar for each data point?
The points which appear to be gains in Si are most likely Si being re-dissolved back into solution, the reason for which is unknown. This may be the Si that is not chemically bonded and part of the hydrogarnet compound but that which has been adsorbed by the monocarboaluminate. This has been added to the text in lines 207-211.
Reviewer 2 Report
The topic of the article is of interest and may represent an essential contribution to the field of desilication of sodium aluminate solutions: nevertheless, the manuscript needs to be improved on accuracy, suitability, and clarity.
1. It is recommended that more references should be added.
2. Please remove irrelevant content and carefully check the full text to avoid similar errors. The content of Line131-149 is like a copy of the Journal Guideline.
3. It is recommended to add the name of the sample for easy reading do not use text 1,2,…
4. Figs. 4 and 5 show incorrect symbols format for temperature.
5. Please explain in detail the reasons for the poor desilication effect of Ca(OH)2
Author Response
1. It is recommended that more references should be added.
Although few, these references represent adequate knowledge for this study and additionally few studies have been done on this topic.
2. Please remove irrelevant content and carefully check the full text to avoid similar errors. The content of Line131-149 is like a copy of the Journal Guideline.
Apologies, the content has been deleted.
3. It is recommended to add the name of the sample for easy reading do not use text 1,2,…
I assume you refer to the slag sample leached to provide the solution for each test. As the focus was on the desilication and not the leaching it was more appropriate to label them by ‘test’.
4. Figs. 4 and 5 show incorrect symbols format for temperature.
Using Excel, it did not allow for the proper symbol in the legend column. Hence the symbol seen.
5. Please explain in detail the reasons for the poor desilication effect of Ca(OH)2
The desilication by Ca(OH)2 cannot be described as poor but in comparison to CaO it is a slower reaction and co-precipitates more Al, this is mentioned in lines 278-279.
Reviewer 3 Report
Review
Dear Authors,
The manuscript entitled „Desilication of sodium aluminate solutions from the alkaline leaching of calcium-aluminate slags” presents evaluation of the desilication of sodium aluminate solutions derived from the leaching of calcium-aluminate slags with sodium carbonate (using fine particles of CaO, Ca(OH)2 and MgO). According to the Authors the aim of this paper is to optimize the use of the desilication process as part of the Pedersen process. The subject is interesting and the paper itself is written in correct scientific language. The data were selected and presented correctly and the results meet the purpose of the paper. The only drawback is that there are too few references to the literature items, especially in Chapter 4 "Discussion", with the result that the chapter is more a description/presentation of one's own test results, rather than relating them to test results carried out previously and described in papers by other researchers. My overall recommendations is to accept this manuscript after minor revision (text correction and completion). Some minor mistakes can be found below.
Minor mistakes:
- the Keywords (e.g. desilication; alkaline leaching) also appear in the title of this article; it is better to avoid such repetition and change some of the repeated keywords while leaving the title unchanged;
- abstract - please underline the main goal (aim – some short info) of the paper (the aim of this work is given in lines 95-97);
- lines 41, 43 et al.: „Na2O.Al2O3.2SiO2” – „.”?
Best regards,
Reviewer
Round 2
Reviewer 1 Report
Accept in the present form